# A Predictive Model for Abnormal Bone Density in Male Underground Coal Mine Workers

**DOI:** 10.3390/ijerph19159165

**Published:** 2022-07-27

**Authors:** Ziwei Zheng, Yuanyu Chen, Yongzhong Yang, Rui Meng, Zhikang Si, Xuelin Wang, Hui Wang, Jianhui Wu

**Affiliations:** Key Laboratory of Coal Mine Health and Safety of Hebei Province, School of Public Health, North China University of Science and Technology, No. 21 Bohai Avenue, Caofeidian New Town, Tangshan 063210, China; zhengzw@stu.ncst.edu.cn (Z.Z.); chenyuanyu@stu.ncst.edu.cn (Y.C.); yangyz@stu.ncst.edu.cn (Y.Y.); mengrui@stu.ncst.edu.cn (R.M.); sizhikang@stu.ncst.edu.cn (Z.S.); wangxuelin@stu.ncst.edu.cn (X.W.); wanghui@stu.ncst.edu.cn (H.W.)

**Keywords:** male underground coal mine workers, bone density abnormalities, XG Boost

## Abstract

The dark and humid environment of underground coal mines had a detrimental effect on workers’ skeletal health. Optimal risk prediction models can protect the skeletal health of coal miners by identifying those at risk of abnormal bone density as early as possible. A total of 3695 male underground workers who attended occupational health physical examination in a coal mine in Hebei, China, from July to August 2018 were included in this study. The predictor variables were identified through single-factor analysis and literature review. Three prediction models, Logistic Regression, CNN and XG Boost, were developed to evaluate the prediction performance. The training set results showed that the sensitivity of Logistic Regression, XG Boost and CNN models was 74.687, 82.058, 70.620, the specificity was 80.986, 89.448, 91.866, the F1 scores was 0.618, 0.919, 0.740, the Brier scores was 0.153, 0.040, 0.156, and the Calibration-in-the-large was 0.104, 0.020, 0.076, respectively, XG Boost outperformed the other two models. Similar results were obtained for the test set and validation set. A two-by-two comparison of the area under the ROC curve (AUC) of the three models showed that the XG Boost model had the best prediction performance. The XG Boost model had a high application value and outperformed the CNN and Logistic regression models in prediction.

## 1. Introduction

Bone mineral density (BMD), known as bone mineral density, is an important indicator of bone strength, reflects the degree of osteoporosis, and is an important predictor of fracture risk [1]. BMD was classified according to World Health Organization standards: normal bone mass, reduced bone mass, and osteoporosis [2]. Osteoporosis (OP) is a systemic disease characterized by low bone mass and destruction of bone structure, abnormal BMD was the main cause of OP. At present, the incidence of OP had jumped to the 7th in common disease, and more than 200 million people worldwide suffered from osteoporosis [3]. The results of the October 2018 China Osteoporosis Epidemiology Survey showed that the prevalence of OP among people aged 40–49 years in China was 3.2%, including 2.2% for men and 4.3% for women, and the highest incidence among people aged 65 years or older was 32.0%. Globally, one osteoporotic fracture occurs every 3 s, and 50% of first-time osteoporotic patients will have another osteoporotic fracture [4]. Abnormal bone density had become an increasingly serious public health problem due to the accelerated ageing of society’s population, abnormal bone density in a large number of people, and a general lack of awareness of bone density.

The coal industry is one of the economic pillar industries in China. There are a great number of people engaged in coal industry, their health status directly related to the development of China’s coal industry. Some studies had pointed out that the special environment of underground coal mines had a significant effect on bone metabolism of people who worked underground for long years [5]. The vast majority of coal miners are male. They worked in the dark, damp and relatively narrow working environment deep underground for a long time, and exposed to occupational harmful factors such as shifts, their risk factors for abnormal bone density differ from those of the general population. A study by the World Health Organization reported that the health and diseases of the population were caused by a variety of factors, including behavioral lifestyle, environmental factors, biogenetic factors and the quality of health care services, among which behavioral lifestyle was the most important influencing factor, accounting for 60%, and miners mostly had bad habits such as smoking, alcohol consumption and high-salt diet [6], which together led to underground workers’ bone metabolic alterations. Currently, domestic and international studies on BMD abnormalities focused on the risk factors and pathogenesis of BMD decline [7,8,9], and there were fewer studies on its early prevention and risk assessment. In addition, domestic studies on BMD had mainly focused on the elderly and menopausal women [10], with fewer studies on BMD in coal miners. If coal miners at risk of abnormal bone density can be identified early, and changing their own unhealthy lifestyles. The number of abnormal BMD can be effectively reduced and the incidence of OP and osteoporotic fractures can be reduced.

Data Mining is the process of extracting knowledge and information with potential application value from large databases and is a new type of information processing system that had developed rapidly in recent years. Disease risk prediction is a very important task in data mining, which is to take the precondition of multiple pathologies of diseases, select multiple influencing factors of diseases, and use suitable statistical analysis methods to construct models so as to predict the probability of occurrence of certain diseases in groups or individuals with certain characteristics [11]. Commonly used models include Logistic regression, neural networks, decision trees, support vector machine (SVM), and so on. Each of these methods had its own characteristics and had been widely and successfully used in the medical field [12]. In recent years, many scholars had used data mining risk prediction methods in the medical field, and all of them had obtained better results. For example, the heart disease risk prediction model based on Convolutional neural network (CNN) by Jian Wang [13] had high prediction accuracy (89.89%) and can accurately predict the risk of heart disease development; Chao-Wen Tan [14] et al. applied convolutional neural networks to effectively improve the robustness and accuracy of heart sound signal classification, which was expected to be applied to machine-aided auscultation. Sethuraman [15] used a feed-forward neural network for feature selection and reduced the number of attributes to 12, which also helped the prediction model achieve 89.4% training accuracy and 82.2% test accuracy. Heydari et al. [6] compared neural network, SVM, decision tree and Bayesian methods in type II diabetes diagnosis and found that the neural network model had the highest accuracy. Tian-Pei Su [16] developed a diabetes risk prediction model based on the eXtreme Gradient Boosting (XG Boost) algorithm, and found that XG Boost was overall a better fit and more accurate than random forest. Hong-Xia Zhang [17] et al. established a prediction model for type II diabetes based on the XG Boost algorithm, which had a good prediction effect with an accuracy of 86.6%. In addition, most of these models currently developed are for disease risk assessment in the general population, and ignore the special groups in the occupational population. There were a large number of coal miners in China, and their special occupational environment such as high temperature, noise, shift work, and other occupational exposures can cause or affect the development of chronic diseases [18,19,20,21,22]. Therefore, prediction models for BMD abnormalities in the general population are not applicable to coal miners. To improve the quality of life and health status of coal miners, there is an urgent need to develop a new predictive model for the risk of BMD abnormalities in coal miners.

Based on the information from the physical examination data of underground coal mine workers, we developed three bone density abnormality prediction models: Logistic regression, CNN, and XG Boost. Overall, our study includes two contributions.

Based on the data information of 3695 underground coal mine workers’ physical examinations, the risk factors of their BMD abnormalities were screened to provide a basis for the development of early prevention strategies for BMD abnormalities in underground coal mine workers.The XG Boost model had better predictive performance of the three models. The XG Boost model can be used to predict the risk of BMD abnormalities in underground coal mine workers, so as to achieve early prevention of BMD abnormalities in underground coal mine workers.

## 2. Materials and Methods

### 2.1. Study Subjects

A total of 3695 male on-the-job underground workers who participated in occupational health physical examination from July to August 2018 in Gequan and Dongpang mines of Hebei Jizhong Energy were selected for the study. Inclusion criteria: age greater than or equal to 18 years; more than 1 year of service. Exclusion criteria: age greater than or equal to 60 years; those with incomplete information; those with congenital metabolic diseases affecting bone metabolism. All study subjects signed an informed consent form. The study was conducted in accordance with the Declaration of Helsinki and was reviewed and approved by the Ethics Committee of North China University of Technology (approval number: 15006).

### 2.2. Information Collection

Face-to-face questionnaires were administered to the study subjects by uniformly trained investigators, and the information collected included (1) demographic information: age, education level, marital status, BMI, per capita monthly household income, etc.; (2) lifestyle habits: smoking, alcohol consumption, sleep, physical exercise, etc.; (3) physical and laboratory examinations: bone mineral density, blood pressure, blood glucose, lipids, etc.; (4) exposure to occupational hazards: length of service, shift work, work intensity, etc.

### 2.3. Laboratory Tests

Fasting venous blood was collected by the doctors early in the morning from the study subjects. Blood specimens were sent to the hospital’s Laboratory Department for blood biochemistry testing using Myriad Automatic Biochemistry Analyzer (BS-800).

### 2.4. Diagnostic Criteria for Abnormal Bone Density

The heel bone density of underground coal mine workers was measured using a CM-200 ultrasonic bone densitometer (FURONO, Japan). Bone density abnormalities were classified according to WHO standards [2].

Normal bone mass: T ≥ −1; reduced bone mass: −2.5 < T < −1; osteoporosis: T ≤ −2.5.

### 2.5. Variable Definitions

Hypertension: blood pressure: systolic blood pressure ≥ 140 mmHg and/or diastolic blood pressure ≥ 90 mmHg, or a previous history of hypertension and current use of antihypertensive medication was defined as hypertension according to the classification criteria of the Chinese Guidelines for the Prevention and Treatment of Hypertension, 2018 Revised Edition [23].Diabetes: according to the classification criteria for glucose metabolic status in the Chinese guidelines for the prevention and treatment of type 2 diabetes (2017 edition) [24], fasting blood glucose ≥ 7.0 mmol/L, or a previous history of diabetes mellitus currently undergoing diabetes treatment was defined as diabetes.Smoking: according to the WHO definition of several terms for smoking [25], smoking status was classified in this study as never smoked, quit smoking and current smoking. Smoking: smoked at least 1 cigarette per day and smoked continuously for more than 6 months; current smoking: was smoking at the time of this survey; never smoked quit: used to smoke but had stopped smoking for at least 6 months at the time of this survey.Drinking: according to the definition of drinking by the Chinese Center for Disease Control and Prevention [26], drinking was classified in this study as never drinking, having stopped drinking, and now drinking. Never drinking: drinking at least once a week and drinking continuously for more than 6 months; Now drinking: drinking at the time of this survey; Already abstaining: used to drink but had stopped drinking for at least 6 months at the time of this survey.Shift situation: a working hour system that requires 24 h continuous work in the production process and is ensured by one or several teams working in shifts. This study classifies the shift situation into the following three cases, never shift, once shift now not shift, and now shift.Years of shift work: the sum of years of shift work performed, this study divided the years of shift work into 5 groups, 0, 0~, 10~, 20~, and more than 30 years.Exercise: exercise more than three times a week, no less than 30 min each time.Body mass index: BMI = weight (kg)/height^2^ (m^2^). The normal range of body weight is BMI < 24 kg/m^2^, the overweight range is 24.0 kg/m^2^ ≤ BMI < 28.0 kg/m^2^, and the obese range is BMI ≥ 28.0 kg/m^2^.Dyslipidemia: according to the Chinese guidelines for the prevention and treatment of dyslipidemia in adults (revised 2016) [27], serum total cholesterol ≥ 6.2 mmol/L, and/or triglycerides ≥ 2.3 mmol/L, and/or LDL cholesterol ≥ 4.1 mmol/L, and/or HDL cholesterol < 1.0 mmol/L, or a previous A history of hyperlipidemia and current use of lipid-lowering drugs defined as dyslipidemia.High-intensity operations: The physical activity of workers was investigated using the International physical activity questionnaire (IPAQ) (long-volume version) [28], and weekly total physical activity levels ≥ 3000 MET-min/w were defined as high-intensity operations.Medium-intensity work: weekly total force activity level ≥ 600 MET-min/w.

### 2.6. Sample Size Calculation

To ensure that the model could accurately predict the mean of the outcome events, the prevalence of abnormal bone density Ø was reviewed in the literature and was approximately 25% [29], with the margin of error δ set at 0.05, which was calculated to require at least 289 study subjects. As shown in Equation (1).
(1)$$n={\left(\frac{1.96}{\delta }\right)}^{2}\emptyset \left(1-\emptyset \right)$$

To control for the minimum mean error of all individual predicted values, the mean absolute error MAPE was set to 0.05, the expected contraction rate *R*^2^*_CS_* was set to 0.1, and the predictor variable P was approximately 24, which was calculated to require at least 951 study subjects. As shown in Equation (2).
(2)$$n=\exp{\left(\frac{-0.508+0.259ln\left(\emptyset \right)+0.504ln\left(P\right)-ln\left(\mathrm{MAPE}\right)}{0.544}\right)}^{2}$$

To ensure an expected contraction rate of 10% and to reduce model overfitting, *S* was set to 0.1 and the number of study variables P was approximately 24, which was calculated to require at least 2038 study subjects. As shown in Equation (3).
(3)$$n=\frac{P}{\left(S-1\right)ln\left(1-\frac{R_{CS}^{2}}{S}\right)}$$

To ensure the minimum difference between the developed model and the *R*^2^*_CS_* optimization adjustment value, *R*^2^*_CS_* in Equation (4) is 0.1 and max*R*^2^*_CS_* is 0.65, and *S**’* is calculated to be 0.75, which is calculated to require at least 671 study subjects. As shown in Equations (4) and (5).
(4)$$n=\frac{R_{CS}^{2}}{R_{CS}^{2}+\delta maxR_{CS}^{2}}$$
(5)$$n=\frac{P}{\left(S'-1\right)ln\left(1-\frac{R_{CS}^{2}}{S'}\right)}$$

It was calculated that a minimum of 2038 individuals needed to be included. A total of 3695 individuals were included in the study, and the sample size met the needs of the study.

### 2.7. Statistical Methods

Excel 2016 was used to establish the original database, and IBM SPSS24.0 was used for statistical analysis. Count data were described by rate or composition ratio, and χ^2^ test was used for comparison between groups; Ordinal data were described by rate or constituent ratio, and the Kruskal-Wallis test was used for comparison between groups. measurement data obeying normal distribution were described by mean and standard deviation, and non-normally distributed data were expressed as median and quartiles; multi-factor unconditional Logistic was used for multivariate analysis of influencing factors. The test level α = 0.05.

### 2.8. Software and Hardware Platform

The sample data are randomly selected in the ratio of 7:2:1 to divide the training set, test set and validation set. Dataset partition codes as shown in Appendix A.

#### 2.8.1. Logistic Regression Model

The Logistic regression model was built using the sklearn package. The Logistic model codes as shown in Appendix A.

#### 2.8.2. CNN Model

Convolutional neural network mainly consists of input layer, convolutional layer, pooling layer, fully connected layer and output layer. In this study, the CNN model was constructed by numpy package. The input layer included 2 nodes, both hidden layers contained 5 nodes, and the output layer included 1 node. The sigmoid function was used as the excitation function, softmax was used for probability normalization, cross-entropy loss function was used as the loss function, and stochastic gradient descent was used as the optimizer. The model was trained by updating the network parameters according to the computed gradients. The CNN model codes as shown in Appendix A.

#### 2.8.3. XG Boost Model

The XG Boost model codes as shown in Appendix A. Based on the m eigenvalues and the n sample data, the XG boost prediction model was obtained by the following equation:(6)$${\hat{y}}_{i}=\varphi \left(X_{i}\right)=\overset{K}{\underset{k=1}{\sum }}f_{k}\left(X_{i}\right),f_{k}\in F$$

In the formula:

i-number of total physical examination data samples

K-total number of trees

$f_{k}$-the k tree

Once the predicted values were obtained, the objective function was obtained by the following equation.
(7)$$O_{bj}\left(\theta \right)=\overset{n}{\underset{i=1}{\sum }}l\left(y_{i},{\hat{y}}_{i}\right)+\overset{K}{\underset{k=1}{\sum }}\Omega f_{k}$$

In the formula:

$l\left(y_{i},{\hat{y}}_{i}\right)$-the training error of the sample $x_{i}$

$\Omega f_{k}$-the regular term of the k tree
(8)$$\Omega f_{t}=\gamma T+\frac{1}{2}\lambda \overset{T}{\underset{j-1}{\sum }}w_{j}^{2}$$

In the formula:

T-number of leaf nodes

w-score values of leaf nodes

γ-parameter to balance the complexity of the model

λ-Parameter to balance the complexity of the model

XG Boost was an additive model in which the objective function changes each time a tree was added to the model. With an additive strategy, the objective function can be written as:(9)$$O_{bj}\left(\theta \right)=\overset{n}{\underset{i=1}{\sum }}l\left(y_{i},{\hat{y}}_{i}^{\left(t\right)}\right)+\overset{t}{\underset{k=1}{\sum }}\Omega f_{k}=\overset{n}{\underset{i=1}{\sum }}l\left(y_{i},{\hat{y}}_{i}^{\left(t-1\right)}+f_{t}\left(x_{i}\right)\right)+\Omega \left(f_{t}\right)+C$$

A Taylor expansion of the loss function via Equation (9) was viewed as, and the final loss function can be written as:(10)$$f\left(x+\Delta x\right)\approx f\left(x\right)+f^{\prime }\left(x\right)\Delta x+\frac{1}{2}f^{\prime \prime }\left(x\right)\Delta x^{2}$$
(11)$$O_{bj}\left(\theta \right)\approx \overset{n}{\underset{i-1}{\sum }}\left(g_{i}f_{i}\left(x_{i}\right)+\frac{1}{2}h_{i}f_{i}{\left(x_{i}\right)}^{2}\right)+\gamma T+\frac{1}{2}\lambda \overset{T}{\underset{j-1}{\sum }}w_{j}^{2}+C$$

### 2.9. Model Evaluation

The model prediction effect was evaluated comprehensively in terms of both discrimination and calibration. As shown in Table 1.

### 2.10. Quality Control

The surveyors were trained by a unified induction, and the questionnaires were checked three times after recovery, and the entry system was double-entry to ensure the accuracy of the information. Measuring instruments were maintained and regularly calibrated by dedicated personnel. 1~2 workers were randomly selected, and the second measurement was made by the surveyor each day, and the results were compared to ensure the consistency of the measurement results. Ten workers were randomly selected to use CM-200 ultrasonic bone densitometer and QCT bone densitometer for bone density measurement, and the QCT bone densitometer measurement results were used to calibrate the CM-200 ultrasonic bone densitometer to ensure the accuracy of bone density measurement results.

## 3. Results

### 3.1. General Demographic Characteristics

The prevalence of BMD abnormalities was 28.25% among the 3695 male coal mine workers included in the study. Analysis of the age of the study subjects revealed that the age range of the study subjects was (19–59) years, with a mean age of (39.04 ± 8.41) years. The results showed that the differences in the prevalence of BMD abnormalities among male coal mine workers were statistically significant (*p* < 0.05) across age, education level, BMI, marital status, fracture, smoking status, drinking status, exercise and Sleep time (h), and not statistically significant (*p* > 0.05) across income, diabetes, hypertension and dyslipidemia groups. As shown in Table 2.

### 3.2. Analysis of Occupational Hazardous Factors and Prevalence of Bone Density Abnormalities

The analysis of occupational hazardous factors in 3695 workers showed that the prevalence of BMD abnormalities in workers increased with working ages and shift length. There were statistically significant differences (*p* < 0.05) between groups of working ages, shift conditions, shift length, high intensity work and medium intensity work. As shown in Table 3.

### 3.3. Logistic Regression Analysis of Risk Factors for Abnormal Bone Density

It had been reported in the literature [30] that hypertension and diabetes had an effect on BMD, so these indicators were also included in the model building. The variable assignments are shown in Table 4.

All independent variables were tested for collinearity diagnostics and found to be free of multicollinearity, as shown in Table 5.

The results of the multifactorial analysis showed that age, low level of education, diabetes, hypertension, fractures, smoking status, drinking status, shift situation, high intensity work and medium intensity work were all risk factors for abnormal bone mineral density, with BMI, physical activity and Sleep time (h) as protective factors. As shown in Table 6.

### 3.4. Bone Density Abnormalities Models Construction and Evaluation

Based on the results of the multifactorial analysis of factors, we included in the model 13 independent variables that were significant for the multifactorial analysis, including age, education, BMI, diabetes, hypertension, fracture, smoking, alcohol consumption, shift work status, heavy workload, moderate workload, exercise, and sleep duration. The sample data were partitioned, with 70% of the training set, 20% of the test set and 10% of the validation set, to construct Logistic, CNN and XG Boost models.

The results of training set sample of 2586 cases (70%) showed that the sensitivity, Youden index, F1 score, AUC (95% CI), Brier score, Log loss, and Calibration-in-the-large of the XG Boost model were 82.058%, 0.715, 0.919, 0.858 (0.839~0.876), 0.040, 0.147, and 0.020, better than the other two models. The CNN model had a better specificity of 91.866%. The Logistic regression model performed worse. As shown in Table 7.

The results of test set sample of 739 cases (20%) showed that the XG Boost model had a sensitivity, specificity, Youden index, F1 score, AUC (95% CI), Brier score, Log loss, and Calibration-in-the-large of 76.555%, 88.302%, 0.649, 0.753, and 0.824(0.787~0.861), 0.107, 0.358, and 0.019, better than the other two models. As shown in Table 7.

The results of test set sample of 370 cases (10%) showed that the sensitivity, specificity, Youden index, F1 score, AUC (95% CI), Brier score, Log loss, and Calibration-in-the-large of the XG Boost model were 76.555%, 85.827%, 0.626, 0.787, and 0.813 (0.762~0.864), 0.040, 0.147, and 0.019, better than the other two models. As shown in Table 7.

A two-by-two comparison of the area under the ROC curves (AUC) of the three models showed that the XG Boost model had the best prediction performance, followed by the CNN model, and the Logistic model had the worst prediction performance, with the differences being statistically significant (*p* < 0.017). The test set results showed that the XG Boost model had the best prediction performance, and the differences were all statistically significant (*p* < 0.017). The results of the validation set showed that the XG Boost model outperformed the CNN model, and the differences were statistically significant (*p* < 0.017); the results of the test and validation sets showed that the differences in prediction performance between the Logistic and CNN models were not statistically significant (*p* > 0.017). AS shown in Table 8 and Figure 1.

The XG Boost model Brier Score, Log Loss, and Calibration-in-the-large metrics all outperformed the CNN and Logistic regression models. The calibration curves for the training, test and validation sets were all close to the diagonal, with no serious deviations in the results, and the calibration curves for the Logistic regression model were more deviant. As show in Figure 2a–c.

In summary, the XG Boost model had excellent performance in the training set, test set and validation set, its predicted risk was in good agreement with the actual occurrence of risk, and each evaluation indexes were significantly better than the CNN model and the Logistic regression model. The XG boost model was the optimal model for this study.

## 4. Discussion

Building risk prediction models was important for early identification and intervention of diseases. Early detection, diagnosis and treatment can contribute to tertiary prevention strategies for the disease. Machine learning had shown advantages in disease models. Meng D et al. conducted a machine learning study on the incidence of hand, foot and mouth disease in all provinces of mainland China and found that the predictive ability of the XG Boost model was generally better than that of the random forest model [31]. Li Z et al. constructed a novel predictive model integrating GCN, CNN and squeeze inspired network (GCSENet) for identifying miRNA-disease associations. By applying the three models together, Li Z et al. obtained an AUC of 0.950 and an F1 score of 0.864, which satisfactorily predicted miRNA disease relevance [32]. Workers working underground in coal mines were susceptible to the effects of their environment on bone metabolism [33], and the factors affecting abnormal bone density were different from those in the general population. Therefore, early identification of high-risk groups and strict control of their influencing factors can reduce the incidence of BMD abnormalities. In this study, 3695 male underground coal mine workers were investigated and their BMD abnormalities were found to be influenced by various factors. By constructing Logistic regression, CNN and XG boost risk prediction models, a comparison of the prediction performance of the three models revealed that the XG boost risk prediction model was the best prediction performance model in this study.

The results of this study showed that the rate of abnormal bone density in a coal mine worker was 28.25%. Based on the importance of the predictor variables in the three models developed, we found that the top four variables were age, BMI, shift work and sleep duration, indicating that these four factors play a very important role in the occurrence of BMD abnormalities. In this study, advanced age was found to be a risk factor for BMD abnormalities, which was consistent with previous findings [34]. The reasons for this might be: decrease in estrogen production with age, which in turn affected parathyroid hormone levels, affecting bone reconstruction and loss of bone mass; the level of secretion decreases with age, inducing increased osteoclast activity and reduced bone content. However, due to the limitation of research conditions, retired workers over 60 years of age were not included in the scope of the study, which may lead to deviations in the research results. Subsequent research can be conducted on retired workers to evaluate the effect of age on bone mineral density of coal mine workers. The detection rate of abnormal bone density was lowest when BMI was between 24.0 and 27.9, which was consistent with previous studies [35]. BMI affected bone density probably because: adipose tissue increased the body’s estrogen content, which favored bone formation; the increased mechanical load on the skeleton at higher BMI can promote bone formation. The risk of abnormal bone density was 1.356 times higher for shift workers compared to workers without shift work. Possible reasons for this were: shift work broke the normal work and rest schedule of workers, making them prone to circadian rhythm disorders and metabolic disorders, and shift work was also associated with the development of diseases such as sleep, hypertension, diabetes and obesity, which can indirectly affect bone density [36]. In terms of sleep duration, this study found that the risk of abnormal BMD in the study participants in the ≥8 h sleep group was 0.561 times lower than that in the <7 h sleep group, which was consistent with the results of previous studies [37]. This may be due to the reduced sleep time of workers, especially at night due to shift work, and exposure to light at night, which reduced melatonin secretion, which can reduce bone mass and also disrupted body metabolism, which in turn affected BMD [38]. The present study found that physical activity was a protective factor for abnormal BMD in male coal mine workers with an OR of 0.725 (95% CI: 0.595, 0.882), similar to the findings of Anupama DS et al. [39]. Hauger AV et al. found that high physical activity was positively associated with total hip bone mineral density compared to a sedentary lifestyle [40]. The present study found that the risk of BMD occurrence was lower in workers with high intensity work than in workers with moderate intensity work, which may be due to skeletal muscle contraction activating bone biomodulation mechanisms that enhance BMD to adapt to exercise load [41]. The results of numerous studies had concluded that smoking had a negative impact on human bone. The results of the present study showed that the risk of abnormal BMD was 1.908 (95CI: 1.547, 2.353) times higher in workers who smoked compared to those who never smoked, which was consistent with the results of previous studies [42,43,44]. The possible reasons for this were that tobacco can affect the production and metabolism of estrogen and androgen, affect the activity of osteoblasts and osteoclasts, and inhibit the vitamin D-parathyroid hormone axis, which can had a negative impact on bones [45]. Adequate vitamin D increases intestinal calcium absorption, promotes bone mineralization, maintains muscle strength, improves balance, and reduces the risk of falls. Vitamin D deficiency can lead to secondary hyperparathyroidism, which increases bone resorption and thus causes or exacerbates osteoporosis. Concurrent calcium and vitamin D supplementation may reduce the risk of osteoporotic fractures. Vitamin D insufficiency also affects the efficacy of other anti-osteoporosis drugs [46]. The current study found that the risk of BMD abnormalities in workers who consumed alcohol was 2.182 (95% CI: 1.684, 2.827) times higher than in non-drinkers, suggesting that alcohol consumption was a risk factor for BMD abnormalities, which was consistent with the findings of some studies [47]. However, some studies had also suggested that moderate alcohol consumption was a protective factor for BMD abnormalities and that it was heavy alcohol consumption that led to BMD abnormalities [48]. Previous studies have been inconsistent regarding the relationship between education level and BMD, and Yan Ren’s analysis of hip fracture risk factors in middle-aged and elderly Chinese found that elderly people with lower education levels were at high risk of fracture [49]. In contrast, Chen et al. [50] observed a positive association between education level and the risk of hip fracture in postmenopausal women in Taiwan. The current study found a high prevalence of BMD in workers with higher education levels. The relationship between education level and BMD needs to be further explored. No consensus conclusion had been reached regarding the effect of alcohol consumption on BMD, which may be related to factors such as study population selection, alcohol intake, frequency of alcohol consumption and type of alcohol consumption.

Logistic regression models were widely used in the field of risk factor screening and disease prediction. It was easy to use and had clear parameter meanings, but the predictive power of Logistic regression models decreased when the data do not meet the requirements [51]. Yan X et al. applied Logistic regression to build an osteoporosis risk model and performed internal and external validation, and the results showed that the C-index was 0.947 for internal validation and 0.946 for external validation, and the calibration curve showed a good agreement between predicted and actual probabilities [52]. In this study, there were limitations in applying the Logistic regression model to BMD prediction for underground workers in coal mines, and all three data sets performed poorly in terms of calibration index, suggesting that the consistency between the predicted and actual results of the Logistic regression model was not high, and that it was prone to bias when used for BMD risk prediction. The CNN model was a deep learning method of multi-layered networks, including input layer, convolutional layer, pooling layer, fully connected layer and output layer. The number of network parameters was effectively reduced and the computational complexity was greatly reduced. It had been used as a neural network model to predict the risk of various diseases in recent years [53,54], but the prediction effect of CNN on different diseases was unstable. For example, Dai G et al. used CNN model to explore the effect of hypertension on the retinal microvascular system, and the results were not satisfactory, with a sensitivity of 60.94%, a specificity of 51.54% and an AUC of 0.6506, which may be due to the fact that the model construction needed to be further improved [55]. Jiang J et al. used a CNN model to identify the degree of left atrial enlargement and showed that the AUCs for normal, mild and moderately severe left atrial enlargement ECGs were 0.942, 0.951 and 0.998 respectively [56]. In this study, the CNN model performed better than the Logistic regression model in terms of calibration metrics, but its ability to distinguish between abnormal and non-abnormal BMD was inferior to the XG Boost model and was not the preferred choice for predicting the risk of abnormal BMD in underground coal mine workers. XG Boost was an improvement of the boosting algorithm based on GBDT, which performed a second-order loss function Taylor expansion, which allowed for higher accuracy. The inclusion of a regular direction in the objective function made the trained model simpler and can effectively combat over-fitting [57]. In this study, the XG Boost model not only had a high ability to distinguish between BMD anomalies and non-anomalies but also had the highest agreement between the prediction results and the actual results, and had the best fit with the BMD anomaly data information of underground workers in coal mines, which can be used for the prediction of BMD anomalies of underground workers in coal mines.

In addition, there were limitations in this study. Our study did not measure vitamin and D levels in underground coal mine workers, did not investigate the type and location of fractures, and did not take into account the medication status of the study subjects. this study only established and completed the internal validation of the risk prediction model for abnormal bone density in coal miners, and no external validation was conducted. There were many methods and sites of bone density measurement, and only one was selected for this study. Moreover, this study was a cross-sectional study, and only information on the prevalence of BMD abnormalities in coal miners was obtained, and the causality argument was less effective, and further cohort studies could be conducted for discussion.

## 5. Conclusions

In this study, the data related to abnormal BMD in male underground coal mine workers were analyzed and found that age ≥ 30 years, Junior secondary school or lower, diabetes, hypertension, fracture, smoking, drinking, shift work, BMI ≥ 28 kg/m^2^, high intensity work and medium intensity work were risk factors. Exercise and sleep time ≥ 7 h were protective factors for bone density abnormalities.

The XG Boost model outperformed the CNN and Logistic regression models in prediction.

## Figures and Tables

**Figure 1 ijerph-19-09165-f001:**
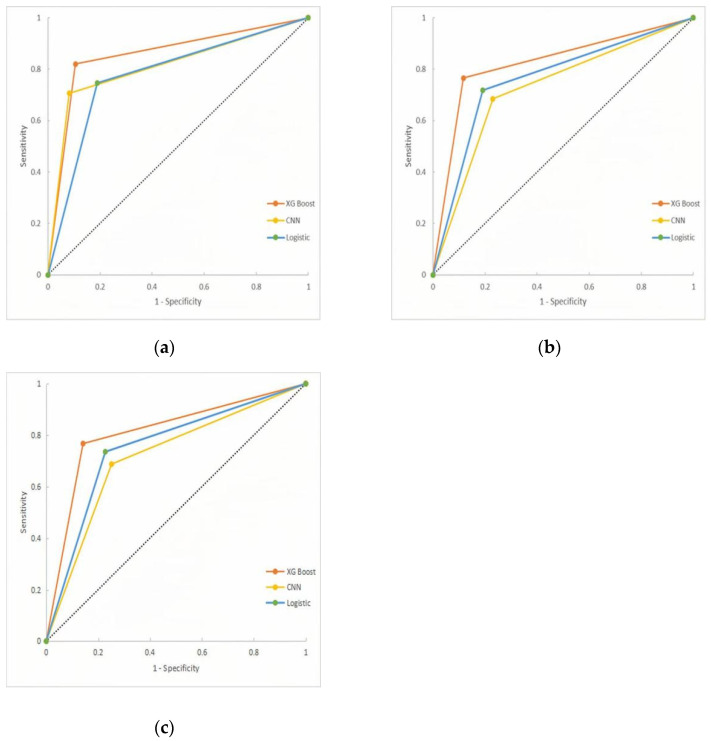
ROC curves of three models (**a**) Training set; (**b**) Test set; (**c**) Validation set.

**Figure 2 ijerph-19-09165-f002:**
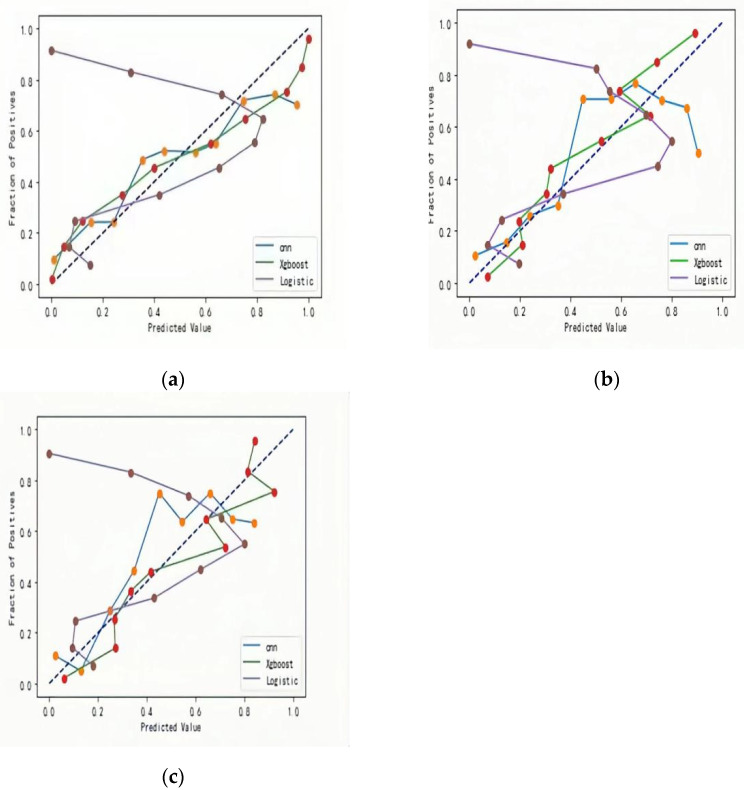
Calibration curves of three models (**a**) Training set; (**b**) Test set; (**c**) Validation set.

**Table 1 ijerph-19-09165-t001:** Model evaluation indexes.

Indicators	Meaning
Sensitivity	The percentage of study participants who actually had BMD and were accurately determined to have BMD by the risk prediction model.
Specificity	The percentage of study participants who did not actually have BMD and were accurately determined to not have BMD by the risk prediction model.
Youden index	Correctness Index, the model correctly determined the total capacity of BMD patients and non-patients.
F1 score	The adjusted mean values of precision and recall, used to evaluate the comprehensive performance of the model.
AUC	Area under the ROC curves.
Brier score	The quantitative score of the model calibration, ranging from 0 to 0.25, the smaller the value, the better the calibration of the model.
Log loss	The error between the true value of the response and the predicted value of the model.
Calibration-in-the-large	The intercept of the calibration curve.

**Table 2 ijerph-19-09165-t002:** General situation of male workers in coal mines.

General Information	Category	Number	Abnormal Bone Mineral Density		χ^2^/H(K)	*p*
			Number	Prevalence Rate (%)		
Age	<30	419	30	7.160	447.518 *	<0.001
	30~	1682	303	18.014		
	40~	945	354	37.460		
	50~	649	357	55.008		
Education level	Junior secondary school or lower	1647	507	30.783	14.956	0.001
	High school and secondary school	1083	308	28.440		
	College and above	965	229	23.731		
BMI (kg/m^2^)	≤23.9	1321	516	39.061	180.677 *	<0.001
	24.0~	1470	411	27.959		
	28.0~	904	117	12.942		
Marital Status	Unmarried	126	24	19.048	6.312	0.043
	Married	3445	980	28.447		
	Other	124	40	32.258		
Family per capita monthly income (Yuan)	<1000	432	124	28.704	4.526 *	0.104
	1000~	2947	847	28.741		
	3000~	316	73	23.101		
Hypertension	No	2379	648	27.238	3.402	0.065
	Yes	1316	396	30.091		
Diabetes	No	3531	990	28.037	1.848	0.174
	Yes	164	54	32.927		
Dyslipidemia	No	2685	748	27.858	0.760	0.383
	Yes	1010	296	29.307		
Fracture	No	2817	694	24.636	76.564	<0.001
	Yes	878	350	39.863		
Smoking status	No smoking	1460	301	20.616	69.733	<0.001
	Quit smoking	245	78	31.837		
	smoking	1990	665	33.417		
Drinking status	No drinking	724	148	20.442	32.053	<0.001
	Alcohol withdrawal	164	37	22.561		
	Drinking	2807	859	30.602		
Exercise	No	1574	501	31.830	17.291	<0.001
	Yes	2121	543	25.601		
Sleep time (h)	<7	1099	419	38.126	136.080 *	<0.001
	7~	1236	387	31.311		
	8~	1360	238	17.500		

* The K-W test was used for ordinal data.

**Table 3 ijerph-19-09165-t003:** Analysis of occupational exposure characteristics of male workers in coal underground.

General Information	Category	Number	Abnormal Bone Mineral Density		χ^2^/H(K)	*p*
			Number	Prevalence Rate (%)		
Working ages	<10	1089	216	19.835	122.264 *	<0.001
	10~	1652	434	26.271		
	20~	539	216	40.074		
	30~	415	178	42.892		
Shift situations	Never	1341	204	15.213	254.064	<0.001
	Once	547	114	20.841		
	Now	1807	726	40.177		
Shift length	0	1341	204	15.213	228.767 *	<0.001
	<10	1098	384	34.973		
	10~	867	262	30.219		
	20~	229	111	48.472		
	30~	160	83	51.875		
High intensity work	No	1502	356	23.702	25.876	<0.001
	Yes	2193	688	31.373		
Medium intensity work	No	437	70	16.018	36.606	<0.001
	Yes	3258	974	29.896		

* The K-W test was used for ordinal data.

**Table 4 ijerph-19-09165-t004:** Assignment table for variables.

Variable	Variable Meaning	Assignment Method
Y	Bone mineral density	1 = normal, 2 = abnormal
X_1_	Age	1 = <30; 2 = 30~; 3 = 40~; 4 = ≥50
X_2_	Educational level	1 = Junior secondary school or lower; 2 = High school and secondary school; 3 = College and above
X_3_	BMI (kg/m^2^)	1= ≤23.9; 2 = 24.0~; 3 = ≥28.0
X_4_	Marital Status	1 = Unmarried; 2 = Married; 3 = Other
X_5_	Hypertension	1 = No; 2 = Yes
X_6_	Diabetes	1 = No; 2 = Yes
X_7_	Fracture	1 = No; 2 = Yes
X_8_	Smoking status	1 = No smoking; 2 = Quit smoking; 3 = smoking
X_9_	Drinking status	1 = No drinking; 2 = Alcohol withdrawal; 3 = Drinking
X_10_	Exercise	1 = No; 2 = Yes
X_11_	Sleep time (h)	1 = <7; 2 = 7~; 3 = ≥8
X_12_	Working age	1 = <10; 2 = 10~; 3 = 20~; 4 = ≥30
X_13_	Shift situation	1 = Never; 2 = Once; 3 = Now
X_14_	Shift length	1 = 0; 2 = <10; 3 = 10~; 4 = 20~; 5 = ≥30
X_15_	High intensity work	1 = No, 2 = Yes
X_16_	Medium intensity work	1 = No, 2 = Yes

**Table 5 ijerph-19-09165-t005:** Multicollinearity of independent variables.

Variable	Tolerance	VIF
Age	0.402	2.489
Educational level	0.806	1.241
BMI (kg/m^2^)	0.879	1.138
Marital Status	0.945	1.058
Hypertension	0.899	1.112
Diabetes	0.937	1.067
Fracture	0.985	1.016
Smoking status	0.938	1.066
Drinking status	0.962	1.039
Exercise	0.932	1.073
Sleep time (h)	0.930	1.075
Working age	0.331	3.021
Shift situation	0.324	3.090
Shift length	0.279	3.589
High intensity work	0.890	1.124
Medium intensity work	0.908	1.101

**Table 6 ijerph-19-09165-t006:** Logistic regression analysis of the influencing factors of abnormal bone mineral density.

Variable	B	S.E	Wald	*p*	OR	95% CI for OR	
						Lower	Upper
Age							
<30			353.284	<0.001			
30~	2.272	0.261	75.697	<0.001	9.699	5.814	16.182
40~	4.145	0.299	191.881	<0.001	63.096	35.101	113.419
≥50	5.779	0.349	273.603	<0.001	323.510	163.112	641.635
Education level							
College and above			32.054	<0.001			
High school and secondary school	0.222	0.117	3.594	0.058	1.249	0.993	1.571
Junior secondary school or lower	0.775	0.137	32.003	<0.001	2.171	1.660	2.841
BMI (kg/m^2^)							
≤23.9			353.668	<0.001			
24.0~	−1.784	0.124	207.232	<0.001	0.168	0.132	0.214
≥28.0	−2.978	0.165	324.298	<0.001	0.051	0.037	0.070
Hypertension	0.243	0.105	5.327	0.021	1.275	1.037	1.567
Diabetes	0.502	0.225	4.990	0.025	1.652	1.064	2.567
Fractures	0.736	0.109	45.982	<0.001	2.087	1.687	2.582
Smoking status							
No smoking			37.743	<0.001			
Quit smoking	0.601	0.194	9.620	0.002	1.825	1.248	2.668
Smoking	0.646	0.107	36.522	<0.001	1.908	1.547	2.353
Drinking status							
No drinking			43.725	<0.001			
Alcohol withdrawal	−0.136	0.277	0.243	0.622	0.872	0.507	1.501
Drinking	0.780	0.132	34.883	<0.001	2.182	1.684	2.827
Exercise	−0.322	0.100	10.294	0.001	0.725	0.595	0.882
Sleep time (h)							
<7			89.013	<0.001			
7~	−0.242	0.114	4.506	0.034	0.785	0.628	0.982
≥8	−1.159	0.128	82.117	<0.001	0.314	0.244	0.403
Shift situation							
Never			181.498	<0.001			
Once	−0.663	0.319	4.314	0.038	0.516	0.276	0.963
Now	1.356	0.271	25.079	<0.001	3.879	2.282	6.593
High intensity work	0.600	0.107	31.590	<0.001	1.822	1.478	2.245
Medium intensity work	1.020	0.176	33.715	<0.001	2.774	1.966	3.915
Constant quantity	−6.318	0.430	215.395	<0.001	0.002	-	-

**Table 7 ijerph-19-09165-t007:** Evaluation of three risk models.

Evaluation Indicator	Training Set			Test Set			Validation Set		
	Logistic	XG Boost	CNN	Logistic	XG Boost	CNN	Logistic	XG Boost	CNN
Sensitivity (%)	74.687	82.058	70.620	71.749	76.555	68.447	73.529	76.724	68.750
Specificity (%)	80.986	89.448	91.866	80.814	88.302	76.923	77.239	85.827	74.818
Youden index	0.557	0.715	0.625	0.526	0.649	0.454	0.508	0.626	0.436
F1 Score	0.618	0.919	0.740	0.631	0.753	0.571	0.583	0.787	0.600
AUC (95% CI)	0.778 (0.757~0.799)	0.858 (0.839~0.876)	0.812 (0.792~0.833)	0.763 (0.723~0.802)	0.824 (0.787~0.861)	0.727 (0.685~0.769)	0.754 (0.696~0.811)	0.813 (0.762~0.864)	0.718 (0.656~0.779)
Brier Score	0.153	0.040	0.156	0.333	0.107	0.172	0.153	0.040	0.156
Log Loss	0.540	0.147	0.492	1.124	0.358	0.538	0.540	0.147	0.494
Calibration-in-the-large	0.104	0.020	0.076	0.104	0.019	0.071	0.146	0.019	0.077

**Table 8 ijerph-19-09165-t008:** AUC comparison of three models.

Data Set	Model	Difference Value of AUC	SE	95% CI		χ^2^	*p*
				Lower	Upper		
training set	Logistic and XG Boost	0.071	0.008	0.055	0.087	8.715	0.001
	Logistic and CNN	0.019	0.007	0.006	0.032	2.886	0.004
	XG Boost and CNN	0.052	0.008	0.036	0.068	6.256	0.001
test set	Logistic and XG Boost	0.074	0.016	0.042	0.106	4.545	<0.001
	Logistic and CNN	0.022	0.019	−0.016	0.060	1.154	0.248
	XG Boost and CNN	0.096	0.020	0.058	0.135	4.923	<0.001
validation set	Logistic and XG Boost	0.039	0.025	−0.009	0.088	1.580	0.114
	Logistic and CNN	0.047	0.022	0.003	0.090	2.110	0.035
	XG Boost and CNN	0.086	0.020	0.047	0.125	4.281	<0.001

## Data Availability

Data available on request due to restrictions privacy. The data presented in this study are available on request from the corresponding author. The data are not publicly available due to the data are not readily available.

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
