# Peer review of "A Predictive Model for Abnormal Bone Density in Male Underground Coal Mine Workers"

_ijerph, 2022, doi:10.3390/ijerph19159165_

Round 1

Reviewer 1 Report

The paper presents an interesting public health problem related to three prediction models (Logistic regression, CNN, and XG Boost) developed to identify coal mine workers “at risk of abnormal bone density as early as possible”.

Based on the results, the authors consider the XG Boost model to be the one with better performance than the other two. However, this conclusion does not appear to be sufficiently justified by the analysis made.

The authors do not give indications on the variables used for the CNN model and XG Boost model. Therefore, the authors seem to use the logistic regression model to select the variables used to evaluate the performance of the other two models. In this way, they seem to validate the logistic regression model rather than develop and compare alternative predictive models.

The variables associated with abnormal BMD in miners (age ≥ 30 years, college education and above, diabetes, hypertension, etc.) and which should theoretically guide the identification of those at risk of abnormal bone density, are precisely those selected by the Logistic Regression Model.

In addition, some points need further explanation and are reported below about the structure of the paper:

The results and conclusions of the study are unusually anticipated in the Introduction.

In the section Materials and Methods:

       Inclusion criteria: age greater than or equal to 18 years - exclusion criteria age >=60 (18-59) in contrast with line 218 (19-59).

       The Authors should report an explanation of the categorization of the continuous variables. All factors should be presented in the methods indicating the categorizations used, motivating them (see sleep time or intensity of work);

       severity Levels of hypertension and diabetes are not described;

       the shift Situation and shift length variables are not sufficiently explained and described;

       subjects' drug therapy is not considered;

        Chi-squared test for the trend should be used for ordinal variables (tables 2 and 3);

In the Results:

       while the text correctly speaks of prevalence, the incidence rate is used in Table 2;

       it is not clear how the variables included in the logistic model were selected;

       Logistic regression model (Table 6) does not contain reference categories for high-intensity work and medium-intensity work; moreover, the values of the odds ratio of these variables are not coherent with a hypothesis of the dose dependence of the association between intensity of the job and BMD (high-intensity work OR=1.822; medium intensity work OR= 2.774)

       the characteristics of the XGBoost model and the CNN model are not as detailed as those of the logistic regression model: there is no description of the variables that can be associated with BMD, essential to implement corrective actions for the protection of the health of the worker;

       in the analysis of the collinearity of the variables of the regression model, working-age, shift situation, and shift length variables are identified as highly collinear; this evidence is not sufficiently interpreted and is not explained because the variables shift Situation and shift length are selected instead of working age, which also seems to be very important in the identification of miners at risk of abnormal bone density as early as possible;

       In Table 6 "Exercise" seems to be a mode of Drinking status, rather than a new variable; same for Hypertension, Diabetes, and Fractures;

       Table 7 does not show the values of the AUC but only the confidence intervals at 95%. Instead, it would be useful to insert the values of the AUC to compare better the differences between AUC shown in table 8;

       The AUC values shown in Figure 1 are not consistent with the results of Tables 7 and 8;

       It is not explained whether the two-to-two AUC comparisons are correct for the number of comparisons.

In the Discussion:

       Neither the significance of the Education level variable is adequately discussed, nor is the divergence of results between Table 2 (where a higher educational level corresponds to a lower prevalence of BMD) and Table 6 (where a higher educational level is more associated with BMD).

In the Bibliography:

a citation is repeated twice (6 and 16):

Heydari, F; Rafsanjani, M.K. A Review on Lung Cancer Diagnosis Using Data Mining Algorithms. Current Medical 462 Imaging, 2021, 17, 16-26. doi:10.2174/1573405616666200625153017

Author Response

Dear Reviewer,

Thank you for your valuable comments. I have attached the letter of recommendation,please see the attachment. Thank you again for your outstanding contribution to the improvement of the quality of our articles.

I am honored to have your advice. Good luck with your work and happy life.

Student.Zheng Ziwei

Email:zhengzw@stu.ncst.edu.cn

Reviewer 2 Report

There were a large number of coal miners in China, and their special occupational environment such as high temperature, noise, shift work, and other occupational exposures  can cause or affect the development of chronic diseases. Therefore, prediction  models for BMD abnormalities in the general population are not applicable to coal miners.  To improve the quality of life and health status of coal miners, there is an urgent need to 101 develop a new predictive model for the risk of BMD abnormalities in coal miners. The conclusion of this study is  the data related to abnormal BMD in male underground coal mine workers were analyzed and found tha age≥ 30 years, college education and above, diabetes, hypertension, fracture , smoking ,drinking , shift work, BMI ≥ 28 kg/m 2, high intensity work and medium

 Intensity work were risk factors. Exercise and sleep time ≥ 7h were  protective factors for bone density abnormalities. The XG Boost model outperformed the CNN and Logistic regression models in prediction.

The introduction explains  the problem.
The health problems associated with workers in coal mines.
The authors choose one of these, osteoporosis , and based on the new statistical methodologies try to establish a mode of action.
It is part of what we call precision medicine   Matherial and methods. Individuals older than 60 years are excluded. This group has a higher incidence of the disease, which could be a selection bias. Could the authors explain the solution of this problem?   BMD was measured with ultrasound, which is not the standard procedure since it is not a central measurement. Do the authors have general population data with this technique? Have you conducted studies comparing this technique with central BMD? There is no control group Results The type of fractures and their location should be indicated. Methodological aspects were included in results Discussion The discussion is correct, commenting on the different elements of the result.
One factor that may be important is vitamin D levels, which are probably low in this population and are not clearly discussed.

Author Response

Dear Reviewer,

Thank you for your valuable comments. I have attached the letter of recommendation, please see the attachment. Thank you again for your outstanding contribution to the improvement of the quality of our articles.

I am honored to have your advice. Good luck with your work and happy life.

Student.Zheng Ziwei

Email:zhengzw@stu.ncst.edu.cn

Round 2

Reviewer 1 Report

Dear Author,

thank you for your attention in responding to the previous comments. The article has been well reviewed and I believe the manuscript has been sufficiently improved to be published in IJERPH. However, the statistical test used for ordinal variables should also be indicated in the material and methods section.

Best regards.

Author Response

Thank you very much for your valuable comments, we have added the following to line 207 under subheading 2.7 in the Materials and methods section: “Ordinal data were described by rate or constituent ratio, and the Kruskal-Wallis test was used for comparison between groups.

Reviewer 2 Report

The authors have answered the questions

Author Response

Dear Reviewer,

We sincerely appreciate that you have thoroughly checked our manuscript and provided very useful comments to guide our revision. During this revision, we found and corrected three formatting errors:

  1. The author of the fourth line removed the extra comma;
  2. Deleted the blank line of 185 lines;
  3. The χ2 superscript has been modified in line 206 under the subheading of Materials and methods 2.7.

We would like to thank you again for taking the time to review our manuscript.

Sincerely yours, 

Zheng Ziwei,

Email:zhengzw@stu.ncst.edu.cn